# Sleep quality and associated factors among adult patients with epilepsy attending follow-up care at referral hospitals in Amhara region, Ethiopia

**Sintayehu Simie Tsega**[1]*, **Birhaneselassie Gebeyehu Yazew**[1], **Kennean Mekonnen**[2]

**1** Department of Medical Nursing, School of Nursing, College of Medicine and Health Science, University of Gondar, Gondar, Ethiopia, **2** Department of Emergency and Critical Care, School of Nursing, College of Medicine and Health Science, University of Gondar, Gondar, Ethiopia

* simies952@gmail.com

## Abstract

**Data Availability Statement:** All relevant data are within the paper and its Supporting Information files.

### Background

Globally, epilepsy is the commonest neurological disorder in adults. It has significant health and economic consequences to the affected individuals and the family. There is ample evidence that epileptic patients are at increased risk of poor sleep quality than the general population. However, there is limited evidence on sleep quality among epileptic patients and associated factors in Ethiopia. Therefore, this study investigated the prevalence of poor sleep quality and associated factors among adult patients with epilepsy.

### Method

Institutional based cross-sectional study was conducted among adult epileptic patients attending follow-up care at referral hospitals in the Amhara region. A total of 575 epileptic patients were recruited using a stratified systematic random sampling technique. An interviewer-administered semi-structured questionnaire and record review were used for data collection. To assess sleep quality the pretested Pittsburgh Sleep Quality Index (PSQI) tool was used. A binary logistic regression model was used to assess factors associated with poor sleep quality. Variables with a p-value less than 0.2 in the bivariable binary logistic regression analysis were considered for the multivariable binary logistic regression analysis. In the multivariable binary logistic regression analysis, the Adjusted Odds Ratio (AOR) with the 95% Confidence Interval (CI) were reported to declare the statistical significance and strength of association. Model fitness was assessed using the Hosmer-Lemeshow test and was adequate (p>0.05). Multicollinearity of the independent variables was assessed using the Variance Inflation Factor (VIF) and the mean VIF was less than 10.

### Results

A total of 565 participants were enrolled in the study with a response rate of 98.3%. The prevalence of poor sleep quality among adult epileptic patients was 68.8% [95% CI: 64.8%,

**Funding:** The University of Gondar College of Medicine and Health sciences was the funding source for this study. It was funding for data collection and we acknowledge it.

**Competing interests:** The authors have declared that no competing interests exist.

**Abbreviations:** AEDs, Anti-Epileptic Drugs; AOR, Adjusted Odds Ratio; MARs, Medication Adherence Rating scale; CI, Confidence Interval; HADS-A, Hospital Anxiety, and Depression Scale-anxiety component; HADS-D, Hospital Anxiety, and Depression Scale-depression component; OSS-3, Oslo-3 items social support scale; PSQI, Pittsburgh Sleep Quality Index and SHI: Sleep Hygiene Index.

72.5%]. In the multivariable binary logistic regression, being unable to read and write [AOR = 3.16, 95%CI: 1.53, 6.51], taking polytherapy treatment [AOR = 2.10, 95% CI: 1.37, 3.21], poor medication adherence [AOR = 2.53, 95%CI: 1.02, 6.23] and having poor support [AOR = 2.72, 95%CI: 1.53, 4.82] and moderate social support [AOR = 1.89, 95%CI: 1.05, 3.41] were significantly associated with higher odds of poor sleep quality.

## Conclusion and recommendation

Poor sleep quality is a major public health concern in Ethiopia. The patient's level of education, number of medication use, medication adherence, and social support were found significant predictors of poor sleep quality. These findings highlight improving medication adherence and social support are effective strategies to improve the sleep quality of epileptic patients. Besides, it is better to give special emphasis to those epileptic patients with a low level of education and taking polytherapy to enhance sleep quality.

## Background

Epilepsy is a neurological disorder attributed to the disruption of brain electrical activity characterized by abnormal body movement and sensory disturbances [1]. It is the most common form of chronic neurological disease affecting more than 50 million people worldwide, of whom, 80% were living in Low-and Middle-Income Countries (LMICs) [2]. The vast majority (70%) of new-onset of epilepsy occur in adults [3]. Epilepsy imposes huge physical, emotional and economic consequences [4, 5].

Sleep is a natural reversible sleep-wake cycle and physiological process that is characterized by perceptual disengagement [6]. Sleep quality is measured by how long and how healthy an individual sleeps, it also contains how problematic it is for an individual to fall asleep and how many times a person wakes up throughout the night [7, 8]. Based on the 2017 result of the National Sleep Foundation (NSF) in the United States of America (USA), good sleep quality is indicated by sleeping more time while in bed (at least 85% of the total time), falling asleep within 30 minutes or less, waking up not more than once per night, and being awake for 20 minutes or less after initially falling asleep [9].

Poor sleep quality impairs the quality of life, cognitive function, and emotion of epileptic individuals [10]. Adults with epilepsy experience poorer sleep quality than their peer healthy counterparts, owing to the nocturnal seizures, which results in marked decrements in quality of life [11]. Likewise, poor sleep is a serious health problem in the contemporary world and the burden commonly observed among epileptic patients that could activate seizures [12–16]. Further, it can induce various adverse outcomes among epilepsy patients, including excessive daytime sleepiness and cognitive impairment [17]. Poor sleep quality is also related to daytime dysfunction and depression and negatively impacts the quality of life [18].

Sleep quality influences physical, intellectual, and emotional health, and good quality of sleep is an essential element for epileptic patients but is highly overlooked [6]. Its deprivation weakens physical function, lowers productivity, and can cause a mental health problem [19].

Previous studies identified several factors which have a significant association with poor sleep quality among adult epileptic patients, of them, taking more than one antiepileptic drug and poor medication adherence was the strongest risk factor for poor sleep quality [20, 21]. Currently, in developing countries including Ethiopia, the prevalence of epilepsy has been

increased dramatically over time [22]. Different authors showed that epileptic patients are at increased risk of comorbidities, of which poor sleep quality is the commonest. Poor sleep quality in epileptic patients makes clinical management more complicated. Therefore, understanding the magnitude and associated factors of poor sleep quality could help to enhance the anti-epileptic treatment outcomes. There is a dearth of information on the prevalence and associated factors of sleep quality among adult Epileptic patients in the study area.

Despite poor sleep quality among epileptic patients having significant physical, mental, and economic consequences, there is limited evidence on the prevalence of poor sleep quality and associated factors among epileptic patients in Ethiopia. Though there are studies reported in Ethiopia [23, 24], these findings cannot be generalized to the current study area as there is huge socio-economic difference such as level of education, wealth, health care access, and health information between them. Besides, we incorporated important predictors such as perceived stigma which is identified as a proximal predictor of sleep quality among epileptic patients. Therefore, this study aimed to investigate the prevalence and associated factors of poor sleep quality among epileptic patients who had follow-up care at referral hospitals in the Amhara region. The findings of the study would give a spotlight to mental health experts, clinicians, and other stakeholders to design a strategy to mitigate the problem and the associated consequences. Moreover, the results of this study would provide information for healthcare providers, caregivers; zonal and regional health administrative to give attention to adult Epileptic patients to optimize and improve the quality of life through improving on service quality primarily on sleep quality.

## Methods

### Study design, period, and setting

An institutional-based cross-sectional study was conducted among adult epileptic patients attending follow-up care at referral hospitals in the Amhara region from March 23 to April 23, 2021. Amhara region, the second-most populous region in the Federal Democratic Republic of Ethiopia. There are a total of eight Referral Hospitals, of these five are found in the Northwestern part of the region, namely; Debre Markos, Debre Tabor, University of Gondar comprehensive specialized Referral Hospital, Tibebegion Comprehensive Referral Hospital, and Felegehiwot Comprehensive Referral hospital, whereas Dessie referral hospital and Woldia Referral hospitals are eastern Amhara and Debrebirahan referral hospital in southern Amhara. Among those, four referral hospitals were selected randomly by using a simple random sampling method, those hospitals are Felegehiwot referral hospital, a tertiary hospital found in Bahir Dar which is the capital city of the Amhara National Regional State, located approximately 565 km northwest of Addis Ababa, Debre Markos referral hospital placed at Debre Markos town, 256 km from Bahir-Dar, the capital of Amhara Regional State and located 300 km from Addis Ababa, the capital city of Ethiopia, Dessie referral hospital found at Dessie town 480 km from Bahir Dar, the capital city of Amhara regional state and located 401 km away from Addis Ababa, the capital city of Ethiopia and Debretabor comprehensive specialized referral hospital.

**Source and study population.** Adult epileptic patients attending follow-up care at referral hospitals in the Amhara region were the source of the population whereas adult epileptic patients who were attending follow-up care during the data collection period at selected referral hospitals in the Amhara region were the study population. Patients with epilepsy having at least one month of follow-up at selected referral hospitals were included in the study but those who were severely ill and unable to respond were excluded from the study.

**Sample size determination and sampling technique.** The sample size was determined by using a single population proportion formula for the first objective considering the following assumptions (Z = 1.96, d = 0.05, and P = 65.4%) [25]. For secondary objectives i.e. for factors associated with poor sleep quality, the sample size was calculated for factors like sex and number of therapy using Epi-Info version 7 statistical software. Finally, the sample size we obtained from the first objective was the largest and taken as our final sample. By using the design effect as 1.5 and adding a 10% non-response rate, the final estimated sample size was 575. Eight referral hospitals are giving epilepsy follow-up services in Amhara Region. The study was conducted on those four randomly selected Referral Hospitals of epilepsy follow-up care and therefore there are an averagely of 1361 epilepsy patients during the data collection period. Then from them, 575 participants were proportionally allocated to each selected referral hospital and those participants were interviewed by systematic random sampling technique considering every two individuals. To avoid recycling of data's special marks were used in the chart and supported by verbal confirmation whether to participate or not in the previous four weeks. The sampling interval value was determined by dividing the total adult epilepsy clients who have a follow-up at referral hospitals in the Amhara region (n = 1361) by the sample size, 1361/575. The first individual was selected using a lottery method.

**Measurements and data collection technique.** Data were collected by five BSc nurses. In addition to the principal investigator, two supervisors were responsible for monitoring the data collection process and the data was collected through face-to-face interviews and medical chart review for the one-month duration. Sleep quality was assessed by using the Pittsburgh Sleep Quality Index, a Self-report questioner containing 19 items assessing seven components of sleep over the past month: subjective sleep quality, sleep latency, sleep duration, habitual sleep efficiency, sleep disturbances, daytime dysfunction, and use of sleep medications. Each component is scored (range 0–3). A total global PSQI score ranges from 0 to 21 (26) classifying a global score of $\geq 5$ as poor sleep quality [26]. We used Cronbach's alpha to assess the internal consistency of a set of scale or test items. The value varies between 0 and 1 and that a higher value indicates a higher internal consistency. The general rule of thumb is that a Cronbach's alpha of 0.70 and above is good, 0. 80 and above is better, and 0. 90 and above is best. It is computed by correlating the score for each scale item with the total score for each observation, and then comparing that of the variance for all individual item scores [27];

$$\alpha = \left(\frac{K}{K-1}\right)\left(1 - \sum_{i=0}^{k}\partial yi^2/\partial x2\right)$$

Where;
K refers to the number of the scale item
$\delta yi^2$ refers to the variance with the item i
$\delta^2 x$ refers to the variance associated with the observed total scores
In this study, the internal consistency was found Cronbach's alpha $\alpha = 0.86$.

The Hospital Anxiety and Depression Scale (HADS) was confirmed its validity and reliability among Ethiopian HIV-positive subjects with Cronbach's alpha (0.87) [28]. It is a brief and internationally used self-rating scale with 14 items (seven items for each). In this study Cronbach's alpha was for HADS-A was (0.92), HADS-D (0.90), and for both anxiety and depression (HADS) was 0.95.

Perceived stigma was assessed by the Kilifi Stigma Scale of Epilepsy, which was developed and validated in Kilifi, Kenya, with high internal consistency, Cronbach's $\alpha$ of 0.91. It is a three-point Likert scoring system scored as "not at all" (0), "sometimes" (1), and "always" (2). It has fifteen items and a total score was calculated by the addition of all item scores. The score

above the median value of the data showed the presence of perceived/felt stigma [29, 30]. In this study, its Cronbach's alpha was 0.95.

Medication adherence level was evaluated by using the ten-item Medication Adherence Rating Scale (MARS). It includes questions asking adherence and drug attitude, and the total score is the sum of these questions which is supposed to assess adherence in better quality [31]. In this study Crobanch alpha = 0.79. Social support was assessed by the Oslo-3 items social support scale (Oslo SSS), which has three items with a Likert scale [32]. Its Cronbach's alpha was done (0.94). Sleep Hygiene Index (SHI), assesses the practice of sleep hygiene behaviors. It has 13-item each item is rated on a five-point scale ranging from 0 (never) to 4 (always). An overall score varies from 0 to 52 [33]. Cronbach's alpha was done (0.94).

**Variables and data analysis.** The dependent variable was the quality of sleep (poor/good) and the independent variables were Socio-demographic factors (age, sex, marital status, occupational status, level of income, and educational status), Clinically related factors (drug adherence, duration of treatment, seizure type, frequency of seizure, number of medication taken, and comorbidity), Behavioral factors(substance use (alcohol, chat, cigarette, illicit drugs) or other substances) and psychosocial factors(anxiety, sleep hygiene, depression, perceived stigma, and social support). Data were coded and entered to Epi data version 4.6 statistical software and then exported to STATA version 14 statistical software. For analysis Chi-square assumption was checked for categorical variables to check the presence of statistically significant association with sleep quality and to screen them for the binary logistic regression model. The prevalence of poor sleep quality with the 95% Confidence Interval (CI) was reported. For the associated factors, a binary logistic regression model was fitted. Both bi-variable and multi-variable binary logistic regression analyses were done, and variables with a p-value ≤ 0.2 in the bi-variable binary logistic regression analysis were considered for the multivariable analysis. In the multivariable binary logistic regression analysis, the Adjusted Odds Ratio (AOR) with the 95% CI was reported to declare the statistical significance and strength of association between poor sleep quality and the independent variables. The model fitness was assessed by the Hosmer-Lemeshow test (0.31). Multicollinearity of the independent variables was assessed by using Variance Inflation Factor (VIF) and the mean VIF was 2.64.

## Ethical consideration

The study was approved by the Research Ethical Review Committee of the school of nursing, College Medicine, and Health Sciences on behalf of the University of Gondar review board S/n/164/2013. A formal letter indicating the approval was obtained and submitted to Amhara Referral Hospitals' administrative. Oral informed consent was obtained from each participant and personal identification like name and medical registration numbers were not used to maintain confidentiality.

## Results

### Socio-demographic characteristics of the study participants

A total of 565 participants were enrolled in the study, with a 98.3% response rate. Among the study participants more than half (53.27%) were females, 301(53.27%) were married, 165 (29.20%) attended secondary school and 306(54.16%) were orthodox religion followers. The median (±IQR) age of participants was 35 ±14 years. The median average family monthly income was 1500 ±1000 ETB (Table 1).

**Table 1. Socio-demographic characteristics of adult patients with epilepsy attending follow-up care at referral hospitals in Amhara region, Ethiopia, 2021 (n = 565).**

| Variable | Category | Frequency (n) | Percent (%) |
|---|---|---|---|
| **Sex** | Male | 264 | 46.73 |
| | Female | 301 | 53.27 |
| **Age (in years)** | 18–24 | 73 | 12.92 |
| | 25–34 | 195 | 31.51 |
| | 35–44 | 172 | 30.44 |
| | ≥45 | 125 | 22.12 |
| **Marital status** | Married | 268 | 47.43 |
| | Single | 198 | 35.04 |
| | Divorced | 68 | 12.04 |
| | Widowed | 31 | 5.49 |
| **Educational status** | Unable to read and write | 138 | 24.42 |
| | Primary school | 162 | 28.67 |
| | Secondary school | 165 | 29.20 |
| | College and above | 100 | 17.70 |
| **Occupational status** | Farmer | 80 | 14.16 |
| | Government employee | 84 | 14.87 |
| | Housewife | 110 | 19.47 |
| | Merchant | 80 | 14.16 |
| | Self-employee | 128 | 22.65 |
| | Student | 83 | 14.69 |
| **Religion** | Orthodox | 306 | 54.16 |
| | Muslim | 153 | 27.08 |
| | Protestant | 91 | 16.11 |
| | Others* | 15 | 2.65 |
| **Residence** | Urban | 292 | 51.68 |
| | Rural | 273 | 48.32 |
| **Average family monthly income(ETB)** | ≤1000 | 255 | 45.13 |
| | >1000 | 310 | 54.87 |

** Other* = Catholic, joeva, Adventist.

## Clinical characteristics of the participants

Among study participants, more than two-thirds (61.06%) were used polytherapy anti-epileptic medication, 323(57.17) were used phenytoin, 366(64.78%) had one up to two seizure frequencies per month and 384(67.96%) of adult epileptic patients had good medication adherence (Table 2).

## Behavioral and psychosocial characteristics of the study participants

Among study participants, more than one-third (36.46%) of epileptic patients had poor social support and 42.65% had strong social support. Regarding perceived stigma, about 47.43% were felt perceived stigma (Table 3).

## Pittsburgh Sleep Quality Index (PSQI) subscale score

Among study participants, 227(40.18%) rated their overall sleep quality as fairly good. About 133(23.54%) of the study participants slept 85% and above of their time spent in bed. The

**Table 2. Clinical characteristics of adult patients with epilepsy attending follow-up care at referral hospitals in Amhara region, Ethiopia, 2021 (n = 565).**

| Variable | Category | | Frequency(n) | Percent (%) |
|---|---|---|---|---|
| Therapy | One | | 220 | 38.94 |
| | Two and above | | 345 | 61.06 |
| Medication type | Carbamazepine | Yes | 200 | 35.40 |
| | | No | 365 | 64.60 |
| | Phenobarbital | Yes | 311 | 55.04 |
| | | No | 254 | 44.96 |
| | Phenytoin | Yes | 323 | 57.17 |
| | | No | 242 | 42.83 |
| | Sodium valproate | Yes | 99 | 17.52 |
| | | No | 466 | 82.48 |
| | Other medication | Yes | 16 | 2.83 |
| | | No | 549 | 97.17 |
| Duration of treatment in years | <5 | | 480 | 84.96 |
| | 6–10 | | 78 | 13.80 |
| | ≥11 | | 7 | 1.24 |
| Comorbidity | Yes | | 20 | 3.54 |
| | No | | 545 | 96.46 |
| Seizure frequency/ month | 0 | | 50 | 8.85 |
| | 1–2 | | 366 | 64.78 |
| | ≥3 | | 149 | 26.37 |
| Seizure type | Focal | | 13 | 2.30 |
| | Generalized | | 552 | 97.70 |
| Drug adherence | Good | | 384 | 67.96 |
| | Poor | | 181 | 32.04 |

Other medication = Chlorothiazide and Hydrochlorothiazide.

**Table 3. Behavioral and psychological characteristics of adult patients with epilepsy attending follow up care at referral hospitals in Amhara region, Ethiopia, 2021 (n = 565).**

| Variables | Category | Frequency(n) | Percent (%) |
|---|---|---|---|
| Substance use | | | |
| Alcohol drinking | Yes | 82 | 14.51 |
| | No | 483 | 85.49 |
| Chat chewing | Yes | 6 | 1.06 |
| | No | 559 | 98.94 |
| Cigarette smoking | Yes | 8 | 1.42 |
| | No | 557 | 98.58 |
| Anxiety | no | 285 | 50.44 |
| | yes | 280 | 49.56 |
| Depression | No | 292 | 51.68 |
| | yes | 273 | 48.32 |
| Perceived stigma | No | 297 | 52.57 |
| | yes | 268 | 47.43 |
| Sleep hygiene | Poor | 276 | 48.85 |
| | Good | 289 | 51.15 |

**Table 4. The Pittsburgh Sleep Quality Index (PSQI) subscale scores among adult epileptic patients attending follow-up care at referral hospitals in Amhara region, Ethiopia, 2021.**

| PSQI subscale | Category | Frequency (n) | Percent |
|---|---|---|---|
| **sleep latency (C2)** | <15mints+not during the past month | 222 | 39.29 |
| | 16–30 mints+ less than once a week | 226 | 40.00 |
| | 31-60min+ once or twice a week | 71 | 12.57 |
| | >60 mints + three or more times a week | 46 | 8.14 |
| **Sleep duration (C3)** | >7hrs | 477 | 84.42 |
| | 6–7 hrs | 70 | 12.39 |
| | 5–6 hrs | 11 | 1.95 |
| | <5 hrs | 7 | 1.24 |
| **Habitual sleep efficiency (C4)** | ≥85% | 133 | 23.54 |
| | 75%-84% | 5 | 0.88 |
| | 65%-74% | 35 | 6.19 |
| | <65% | 392 | 69.38 |
| **Sleep disturbance (C5)** | None | 90 | 15.93 |
| | 1–9 | 389 | 68.85 |
| | 10–18 | 58 | 10.27 |
| | 19–27 | 28 | 4.96 |
| **Medication use for sleep (C6)** | Not during the last month | 510 | 90.27 |
| | less than once a week | 28 | 4.96 |
| | once or twice a week | 23 | 4.07 |
| | ≥3 times a week | 4 | 0.71 |
| **Daytime dysfunction (C7)** | No problem | 314 | 55.58 |
| | Slight problem(1-2/week) | 119 | 21.06 |
| | Moderate problem>2/week) | 110 | 19.47 |
| | Big problem >3/week | 22 | 3.89 |
| **Global sleep quality** | | Median | IQR |
| | | 6 | 4 |

**Key: IQR- Inter Quartile Range, PSQI-Pittsburgh Sleep Quality Index, and C-component.

participants went to bed on average at 8:00 pm and wakeup at 7:00 Am. moreover, the mean-time for sleep was 9:00hrs (Table 4).

The most frequent reason for the difficulty in maintaining sleep was pain 334(59.12%) followed by midnight wake up or early morning in 273 patients (48.32%), to use the bathroom in 191 patients (33.81%), unable to fall asleep within 30 minutes in 188 patients (32.27%), having bad dreams in 139 patients (26.60%), coughing in 128 patients (22.65%), feeling of too hot in 121 patients (21.42%), having difficulty of breathing in 119 patients (21.06%), felling of too cold in 114 patients (20.18%) and others (for taking medication and for praying) in 29 patients (4.24%). In this study, the prevalence of poor sleep quality among adult epileptic patients was found to be 68.8% (95%CI; 64.8%, 72.5%). The prevalence of poor sleep quality among adult epileptic patients taking polytherapy treatment was 76.8% (95%CI; 72.0%, 80.1%) whereas monotherapy treatment was 56.4% (95%CI; 49.7%, 62.8%).

## Factors associated with poor sleep quality

In the multivariable binary logistic regression analysis, educational status, number of medications, social support, and medication adherence were significantly associated with poor sleep quality. An epileptic patient who was unable to read and write had 3.16 times (AOR = 3.16,

**Table 5. Factors associated with poor sleep quality among adult epileptic patients attending follow-up care at referral hospitals in Amhara region, Ethiopia, 2021.**

| Variable | Category | Sleep quality | | COR(95%CI) | AOR(95%CI) | P-value |
|---|---|---|---|---|---|---|
| | | Poor (%) | Good (%) | | | |
| **Sex** | Female | 222(73.75) | 79(26.25) | 1.63(1.14,2.34) | 1.26(0.83,1.92) | 0.28 |
| | Male | 167(63.26) | 97(36.74) | 1.00 | 1.00 | |
| **Age in years** | 18–24 | 41(56.16) | 32(43.84) | 1.00 | 1.00 | |
| | 25–34 | 134(68.72) | 61(31.28) | 1.71(0.99,2.98) | 1.51(0.82,2.79) | 0.18 |
| | 35–44 | 123(71.51) | 49(28.49) | 1.96(1.11,3.46) | 1.31(0.69,2.49) | 0.40 |
| | ≥45 | 91(72.80) | 34(27.20) | 2.09(1.14,3.83) | 0.81(0.39,1.68) | 0.57 |
| **Educational status** | Unable to read and write | 121(87.68) | 17(12.32) | 4.18(2.18,8.00) | **3.16(1.53,6.51)**\*\* | 0.002 |
| | Primary school | 92(56.79) | 70(43.21) | 0.77(0.46,1.29) | 0.57(0.32,1.01) | 0.06 |
| | Secondary school | 113(68.48) | 52(31.52) | 1.28(0.76,2.15) | 1.02(0.55,1.86) | 0.96 |
| | College and above | 63(63.00) | 37(37.00) | 1.00 | 1.00 | |
| **Average monthly income(ETB)** | ≤1000 | 191(74.90) | 64(25.10) | 1.69(1.17,2.43) | 1.30(0.84,2.01) | 0.23 |
| | >1000 | 198(63.87) | 112(36.13) | 1.00 | 1.00 | |
| **Therapy** | Two and above | 265(76.81) | 80(23.19) | 2.56(1.78,3.69) | **2.10(1.37,3.21)**\*\* | 0.001 |
| | One | 124(56.36) | 96(43.64) | 1.00 | 1.00 | |
| **Seizure frequency per month** | 0 | 32(64.00) | 18(36.00) | 1.00 | 1.00 | |
| | 1–2 | 234(63.93) | 132(36.07) | 0.99(0.54,1.84) | 0.95(0.46,1.95) | 0.89 |
| | ≥3 | 123(82.55) | 26(17.45 | 2.66(1.30,5.44) | 0.64(0.21,1.98) | 0.44 |
| **Substance use** | Yes | 82(85.42) | 14(14.58) | 3.09(1.69,5.62) | 1.07(0.49,2.34) | 0.85 |
| | No | 307(65.46) | 162(34.54) | 1.00 | 1.00 | |
| **Anxiety** | Yes | 210(75.00) | 70(25.00) | 1.78(1.24,2.55) | 0.64(0.27,1.47) | 0.29 |
| | No | 179(62.81) | 106(37.19 | 1.00 | 1.00 | |
| **Depression** | Yes | 209(76.56) | 64(23.44) | 2.03(1.41,2.93) | 1.94(0.83,4.55) | 0.12 |
| | No | 180(61.64) | 112(38.36) | 1.00 | 1.00 | |
| **Perceived stigma** | Yes | 197(73.51) | 71(26.49) | 1.52(1.06,2.18) | 0.73(0.35,1.51) | 0.39 |
| | No | 192(64.65) | 105(35.35) | 1.00 | 1.00 | |
| **Drug adherence** | poor | 154(85.08) | 27(14.92) | 3.62(2.29,5.71) | **2.53(1.02,6.23)**\* | 0.04 |
| | good | 235(61.20) | 149(38.80) | 1.00 | 1.00 | |
| **Social support** | poor | 169(82.04) | 37(17.96) | 3.96(2.56,6.14) | **2.72(1.53,4.82)**\*\* | 0.001 |
| | moderate | 91(77.12) | 27(22.88) | 2.93(1.78,4.82) | **1.89(1.05,3.41)**\* | 0.03 |
| | strong | 129(53.53) | 112(46.47) | 1.00 | 1.00 | |
| **Sleep hygiene** | poor | 208(75.36) | 68(24.64) | 1.82(1.27,2.62) | 1.14(0.67,1.94) | 0.62 |
| | good | 181(62.63) | 108(37.37) | 1.00 | 1.00 | |

\*p-value<0.05,

\*\*p-value<0.01 COR = Crude Odds Ratio, AOR = Adjusted Odds Ratio and CI = confidence interval. Model fitness: Hosmer Lemeshow Goodness of fittest; P value was 0.31.

95%CI: (1.53, 6.51)) higher odds of poor sleep quality as compared to those who attained college and above. The odds of poor sleep quality among patients who took more than one anti-epileptic drug were 2.10 times (AOR = 2.10, 95%CI: (1.37, 3.21)) higher than those who took one anti-epileptic drug. Epileptic patients who had poor medication adherence were 2.53 times (AOR = 2.53, 95%CI: (1.02, 6.23)) higher odds of poor sleep quality compared to those who had good medication adherence. Epileptic patients who had poor social support had 2.72 times (AOR = 2.72, 95%CI: (1.53, 4.82)) and those who had moderate social support had 1.89 times (AOR = 1.89, 95%CI: (1.05, 3.41)) higher odds of poor sleep quality compared to those who have strong social support (Table 5).

## Discussion

Poor sleep quality is a serious health problem for adult patients with Epilepsy in low-and middle-income countries including Ethiopia. This study found that the prevalence of poor sleep quality among adult Epileptic patients was 68.8%. This finding was in line with studies conducted in Ethiopia, Addis Ababa 65.4% [25], United States of America 72% [34], and Brazil 67.3% [26]. However, it was higher than studies reported in Taiwan 50% [20], Korea 41.1 [35], Spain 53.6% [36], India 48% [37], Turkey 42.7% and Southeast Asia 33% [8]. Possible reasons for the difference may be due to the use of different screening tools and cutoff points.

Besides, the discrepancy may be due to exclusion and inclusion criteria of the study participants (comorbidity, medications known to affect sleep, night work, shift work, and Patients who received phenobarbital) were excluded in the study conducted in Taiwan [20]. Patients having comorbidity, night workers, shift workers were excluded whereas patients having been followed for epilepsy for at least two years were the Inclusion criteria in a study conducted in Turkey [38]. In a study conducted in Korea, epilepsy patients' ages greater than or equal to 20 years were included [35].

In a study done in Karnataka, India patients with comorbid diseases or medications known to affect sleep, other than Valproic acid, and those with substance abuse of any degree were excluded [37].

The finding from this study was lower than a study conducted in Nigeria [39]. The variation may be due to measurement tool difference (Epworth sleepiness score tool was used in a study conducted in Nigeria whereas the PSQI tool was used in this study).

In the current study, associations were observed between poor sleep quality and different independent variables. The odds of developing poor sleep quality among patients who are unable to read and write were nearly three times as compared to those who attended college and above.

The possible justification may be due to the lesser knowledge they may have about seizure-triggering factors and seizure management as well as they may not easily understand instructions given from health professionals and this may result in poor medication adherence, poor seizure control which may result in poor sleep quality. Moreover, it might be also due to fewer employment opportunities because people who are unable to read and write are less likely to have different job offers and less chance of employment by both governmental and non-governmental organizations which may result in stress and turn compromise their sleep quality.

The odds of developing poor sleep quality among patients who took more than one drug were more than two times more likely to develop poor sleep quality as compared to those who took one anti-epileptic drug. This finding is supported by studies conducted in Addis Ababa, Ethiopia Taiwan, India, and Delhi India [20, 25, 40, 41]. The possible reason may be due to that the use of polytherapy anti-epileptic drugs may raise undesirable effects and drug-drug interactions result in poor sleep quality. Even though a few pieces of literature have suggested that in epilepsy, more than one AED in general required in medication-resistant cases, and the higher the number of AEDs needed, the higher is the degree of severity of pharmaco-resistance. therefore the tight relation between poor sleep quality and AED polytherapy suggests that poor sleep quality may arise as a result of the disease itself, or it may be secondary to the effects of medications [34]. Besides this, patients who had Polytherapy medication treatment might have un-affordability issues which result in stress, poor medication adherence, and poor sleep quality.

Epileptic patients who had poor medication adherence to their AED were nearly three times more likely to develop poor sleep quality as compared to their counterparts. This finding was supported by study findings conducted in Addis Ababa, Ethiopia [25], Taiwan [20], India

[41], and Brazil [42]. The possible justification might be that epileptic patients with poor medication adherence to their antiepileptic drug could result in increased seizure frequency, hospital admissions, increased health care cost, and worse clinical outcomes that may contribute patients to having poor sleep quality. Moreover, patients who had poor adherence to their AED may lead to reduced seizure control, decreased work productivity, and seizure-related job loss, stress, and poor sleep quality.

Epileptic patients who had poor Social support were nearly three times and moderate social support were nearly two times more likely to have poor sleep quality as compared to participants who have strong social support. This finding was supported by a study conducted in Southeast Asia [8].

The possible justification might be due to the sense of being not loved and socially lonely can contribute to providing that an uncomfortable environment that hinders the capability to cope with the disease condition results in poor sleep quality.

Moreover, this might be due to persons who had poor social support cannot have the ability to avoid negative feelings in their personal life, not being able to concentrate on their daily activities and they cannot make their life meaningful which may end with poor sleep quality. Furthermore, epileptic patients who have poor social support can contribute to greater perceived stress and hopelessness, they also go through stressful conditions and which consequences in poor sleep quality. Strong social support is believed to endorse good sleep quality by providing a safe context in which close family or friends care for sleepers' enemies or other threats [43].

## Limitation of the study

The study may be prone to recall bias as it assesses sleep conditions before a month. Due to the cross-sectional nature of the study, we are unable to draw the cause-effect relationships of poor sleep quality and the predictors.

## Conclusion

The finding of this study showed that more than two–thirds of the study participants had poor sleep quality at referral hospitals in the Amhara region. Unable to read and write, polytherapy treatment, poor medication adherence, and having poor and moderate social support, were factors that increase the occurrence of poor sleep quality. Therefore, to improve the sleep quality of epileptic patients the health care provider should give special emphasis to those who took multidrug, poorly adhered, or who are unable to read and write through social integration [44].

## Supporting information

**S1 Dataset.**
(DTA)

## Acknowledgments

We would like to thank study participants, data collectors, and supervisors for their unreserved contribution during data collection. Also, we would like to forward our gratitude to Amhara region Referral Hospital administrators, heads of the chronic outpatient department, and health care providers for their valuable support during data collection.

## Author Contributions

**Conceptualization:** Sintayehu Simie Tsega, Birhaneselassie Gebeyehu Yazew, Kennean Mekonnen.

**Data curation:** Sintayehu Simie Tsega, Birhaneselassie Gebeyehu Yazew, Kennean Mekonnen.

**Formal analysis:** Sintayehu Simie Tsega, Birhaneselassie Gebeyehu Yazew, Kennean Mekonnen.

**Funding acquisition:** Sintayehu Simie Tsega, Kennean Mekonnen.

**Investigation:** Sintayehu Simie Tsega, Birhaneselassie Gebeyehu Yazew, Kennean Mekonnen.

**Methodology:** Sintayehu Simie Tsega, Birhaneselassie Gebeyehu Yazew, Kennean Mekonnen.

**Resources:** Kennean Mekonnen.

**Software:** Sintayehu Simie Tsega, Birhaneselassie Gebeyehu Yazew, Kennean Mekonnen.

**Supervision:** Sintayehu Simie Tsega, Birhaneselassie Gebeyehu Yazew, Kennean Mekonnen.

**Validation:** Sintayehu Simie Tsega, Birhaneselassie Gebeyehu Yazew, Kennean Mekonnen.

**Visualization:** Sintayehu Simie Tsega, Birhaneselassie Gebeyehu Yazew, Kennean Mekonnen.

**Writing – original draft:** Sintayehu Simie Tsega, Birhaneselassie Gebeyehu Yazew, Kennean Mekonnen.

**Writing – review & editing:** Sintayehu Simie Tsega, Birhaneselassie Gebeyehu Yazew, Kennean Mekonnen.

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
