## [Decision Letter · Decision Letter 0]

11 Oct 2021

PONE-D-21-26585Sleep quality and associated factors among adult patients with epilepsy attending follow-up care at Referral Hospitals in Amhara Region, EthiopiaPLOS ONE

Dear Dr. Simie,

Thank you for submitting your manuscript to PLOS ONE. After careful consideration, we feel that it has merit but does not fully meet PLOS ONE’s publication criteria as it currently stands. Therefore, we invite you to submit a revised version of the manuscript that addresses the points raised during the review process.

We look forward to receiving your revised manuscript.

Kind regards,

Hamidreza Karimi-Sari, MD

Academic Editor

PLOS ONE

Journal Requirements:

4. PLOS requires an ORCID iD for the corresponding author in Editorial Manager on papers submitted after December 6th, 2016. Please ensure that you have an ORCID iD and that it is validated in Editorial Manager. To do this, go to ‘Update my Information’ (in the upper left-hand corner of the main menu), and click on the Fetch/Validate link next to the ORCID field. This will take you to the ORCID site and allow you to create a new iD or authenticate a pre-existing iD in Editorial Manager. Please see the following video for instructions on linking an ORCID iD to your Editorial Manager account: https://www.youtube.com/watch?v=_xcclfuvtxQ.

5. Thank you for stating the following in the Funding Section of your manuscript:

“The University of Gondar College of Medicine and Health sciences was the funding source for this study.”

We note that you have provided additional information within the Funding. Please note that funding information should not appear in other areas of your manuscript. We will only publish funding information present in the Funding Statement section of the online submission form.

 “The University of Gondar College of Medicine and Health sciences was the funding source for this study. It was funding for data collection and we acknowledge it.”

Reviewers' comments:

Reviewer's Responses to Questions

**Comments to the Author**

1. Is the manuscript technically sound, and do the data support the conclusions?

Reviewer #1: Yes

Reviewer #2: Yes

2. Has the statistical analysis been performed appropriately and rigorously? 

Reviewer #1: Yes

Reviewer #2: Yes

3. Have the authors made all data underlying the findings in their manuscript fully available?

Reviewer #1: No

Reviewer #2: Yes

4. Is the manuscript presented in an intelligible fashion and written in standard English?

Reviewer #1: No

Reviewer #2: Yes

5. Review Comments to the Author

Reviewer #1: The authors have used multiple questionnaires and logistic regression to assess sleep quality in the Amhara Region, Ethiopia. The authors have found that medical adherence, social support, education level and anti-epileptic drug load are significant predictors for sleep quality.

Major points:

1. The language and punctuation throughout the manuscript needs attention. There are many instances where errors make it difficult to interpret the authors meaning. Some examples of the errors are listed below (please note that the errors listed are from the first page but there are many more errors on that first page and throughout the entire manuscript than are listed here):

- Correct neurologic to neurological in the abstract

- Line 52: capitalize first letter at the beginning of the sentence.

- Line 55: incorrect placement of commas

- Line 59: I believe the authors might mean ‘measured’ or ‘gauged’ rather than manifested

- Line 65: replace ‘later’ with ‘after’

- Line 67: poor should be poorer

2. I found the discussion of Cronbach’s alpha confusing and unclear. How was it actually being used and how was it being calculated. Please include the formula for how Cronbach’s alpha is calculated in the paper and explain more clearly how it is being used. For example: Cronbach alpha was calculated for each questionnaire to determine internal consistency, meaning that each question was assessing the same concept. It would also be useful to describe that the value varies between 0 and 1 and that a higher value indicates a higher internal consistency. What is the cut-off value for the questionnaire to be considered ‘valid and reliable’?

3. Sleep comorbidities? Were patients with sleep comorbidities included in this study? If so, please look at how those patients might be influencing the results and include a thorough discussion as to how this might bias your results.

Minor points:

4. Line 239: I assume the patients wakeup at 7AM not PM on average?

5. How does 8pm bedtime compare to healthy individuals in the region? Is that an earlier bedtime than normal?

6. Please separate the discussion of each questionnaire in the methods section into individual paragraphs. It is confusing having some paragraphs discuss one questionnaire and others discuss multiple unrelated questionnaires.

7. Please make sure you are clear for each questionnaire as to how the questions are scored (what range of numbers) and whether a higher/lower score indicates better/worse sleep hygiene (for example).

8. Is the high prevalence of pain causing wake up thought to be related to their epilepsy or is this consistent with the healthy population from the region?

9. Was the ability to read and write associated with medication adherence? Seems with the data at hand you could actually test the theory you are putting forward on lines 295-296.

10. Spelling mistake in title of final figure

Reviewer #2: Important paper that seeks to relate the quality of sleep with sociodemographic, clinical, behavioral and psychosocial factors in patients with epilepsy. Here are suggestions for modification:

In the abstract and in line 87, replace "literature" by "authors" because it is the latter who suggest or show something. There is no need in the summary and discussion to represent the confidence interval. In line 42, replace "common" with important, great presence, any term that stands out more than common.

In the introduction, line 58, improve sleep description. It is not "started" by a circadian cycle but has circadian rhythmicity. Furthermore, the correct one is the phase of wakefulness and not the period. Nor would I say that the quality of sleep "manifests" itself and the term "slumbering" is not very common in the literature.

In line 66, is cited multitude of sequelae, I would like to see this paragraph more developed. The same applies for the consequences of poor sleep quality, on line 71.

I suggest paying attention to some parts of the work regarding the writing of English. For example, the paragraph in line 74 is a science communication text and not a text for a scientific journal at the PLOS ONE level. It should not be abandoned but rewritten with greater scientific rigor. There are also several small grammar errors throughout the text that a review by an English language specialist can be helpful.

In line 83, complete the study description reasoning. On line 91, remove "As to our search of the literature"

I didn't understand the justification in lines 93, 94, 95. Elaborate this part better. On line 96, "Would be able" gets too vague. Need to be more assertive.

In the results, in Table 1: Why age groups are not homogeneous: 18-24; 25-31; 32-38; 39-45; and > 45. I suggest using 5 groups.

In the body of the text, provide more details on the results shown in table 1, 2. Table 2 can be joined with table 3. Rethink the need to include figure 1, or include the other factors that are associated with poor sleep quality, "Being incapable of reading and writing", "undergoing polytherapy", "having poor medication adherence" and "having poor and moderate social support". In table 4, it is not clear "<15mints + not during the last month" (?). Describe in the caption what IQR means. I see no need for Figure 2 unless the results are discussed in the later section.

Remove the subheadings "Subjective sleep quality rate (component one)", "Sleep disturbance (component five)" and "Sleep quality prevalence" Provide a more detailed and joint description of all PSQI results.

Discussion

Do not repeat in line 272 what the objectives of the study are. Add the first paragraphs and remove any mention of standard deviations, confidence intervals, for example (95% CI; 64.8%, 72.5%) because we are in the discussion part and not the results part.

The last paragraph reflects my criticism of the need to revise English. I understood what the authors mean but the text can improve a lot. It is necessary to discuss with references about the outcome of pain (component five). In the conclusions, cite what was said in the summary of the work about how to improve the quality of sleep in these patients.

6. PLOS authors have the option to publish the peer review history of their article (what does this mean?). If published, this will include your full peer review and any attached files.

Reviewer #1: No

Reviewer #2: **Yes: **Leandro Lourenção Duarte

---

## [Author Response · Author response to Decision Letter 0]

16 Nov 2021

Point by point response for editors/reviewers comments 

PLOS ONE Journal 

Manuscript title: Sleep quality and associated factors among adult patients with epilepsy attending follow-up care at Referral Hospitals in Amhara Region, Ethiopia

Manuscript ID: PONE-D-21-26585

Dear editor/reviewer. 

Dear all,

We would like to thank you for this constructive, building, and improvable comments on this manuscript that would improve the content of the manuscript. We considered each comment and clarification questions of editors and reviewers on the manuscript thoroughly. Our point-by-point responses for each comment and question are described in detail on the following pages. Further, the details of changes were shown by track changes in the supplementary document attached.

Response to reviewers’ comment

Reviewer#1

1. The language and punctuation throughout the manuscript needs attention. There are many instances where errors make it difficult to interpret the authors meaning. Some examples of the errors are listed below (please note that the errors listed are from the first page but there are many more errors on that first page and throughout the entire manuscript than are listed here)

-Correct neurologic to neurological in the abstract

- Line 52: capitalize first letter at the beginning of the sentence.

- Line 55: incorrect placement of commas

- Line 59: I believe the authors might mean ‘measured’ or ‘gauged’ rather than manifested

- Line 65: replace ‘later’ with ‘after’

- Line 67: poor should be poorer

Authors’ response: Thank you for reviewer for the comments. We have addressed all the comments. We had extensively edited the entire manuscript for grammatical errors, sentence structure and typographical errors with the help of language experts at the University. (See the revised manuscript)

2. I found the discussion of Cronbach’s alpha confusing and unclear. How was it actually being used and how was it being calculated. Please include the formula for how Cronbach’s alpha is calculated in the paper and explain more clearly how it is being used. For example: Cronbach’s alpha was calculated for each questionnaire to determine internal consistency, meaning that each question was assessing the same concept. It would also be useful to describe that the value varies between 0 and 1 and that a higher value indicates a higher internal consistency. What is the cut-off value for the questionnaire to be considered ‘valid and reliable’?

Authors’ response: Thank you reviewer for the comments. As you said we assessed Cronbach’s alpha to assess the internal consistency, it is ranged from 0 to 1. Literatures suggested that as general rule that α of 0.6 – 0.7 indicates an acceptable level of reliability, and 0.8 and above is considered as a very good level of internal consistency. For further we have included the formula to calculate Cronbach’s alpha and some statement about it. If you need further justification we are ready to provide it. (See the Method section, Page 5-6, line 153-163)

3. Sleep comorbidities? Were patients with sleep comorbidities included in this study? If so, please look at how those patients might be influencing the results and include a thorough discussion as to how this might bias your results. 

Authors’ response: Thank you reviewer for the comments. As you said sleep comorbidities like underlying medical conditions which may have association with sleep quality might influence our findings. And therefore, we considered underlying comorbidity other than epilepsy as independent variable and considered in the analysis. We planned to consider this variable during analysis to control its confounding effect on others, but unfortunately this variable did not fulfill the chi-square assumption /in the bivariable binary logistic regression analysis and we did not considered in the multivariable binary logistic regression analysis. That is why we did not discuss this in the discussion section. (See the revised manuscript)

Minor points

4. Line 239: I assume the patients’ wakeup at 7AM not PM on average?

Authors’ response: Thank you reviewer for the comments. It was editorial error and we have addressed the comment. (See the revised manuscript)

5. How does 8pm bedtime compare to healthy individuals in the region? Is that an earlier bedtime than normal?

Authors’ response: Thank you reviewer for the comment. We have asked this question to compute the component habitual sleep efficiency (PSQI) (Component 4: Habitual sleep efficiency Component 4 (total # of hours asleep) / (total # of hours in bed) x 100 (>85%=0, 75%-84%=1, 65%-74%=2, <65%=3). Simultaneously, we have reported average bedtime, relatively which is usual time in case of Ethiopia. Therefore, if our response does not convince you we are ready to remove this sentence in the manuscript as it is not standalone variable in this study. 

6. Please separate the discussion of each questionnaire in the methods section into individual paragraphs. It is confusing having some paragraphs discuss one questionnaire and others discuss multiple unrelated questionnaires.

Authors’ response: Thank you reviewer for the comment. We accept it and use separate paragraph for each questionnaire. (See Method section, line 146-182, and page 5-6)

7. Please make sure you are clear for each questionnaire as to how the questions are scored (what range of numbers) and whether a higher/lower score indicates better/worse sleep hygiene (for example).

Authors’ response: Thank you for the comments. We have computed each scores based on the tools recommendation and we make sure that we have checked the scores are correct. (See the revised manuscript)

8. Is the high prevalence of pain causing wake up thought to be related to their epilepsy or is this consistent with the healthy population from the region?

Authors’ response: Thank you reviewer for the comment. To compute Component 5, there is one question about sleep disturbance and cause of disturbance. One of the option was pain and the high prevalence of pain causing wake up was high and we believe that this pain is related to epilepsy and its associated morbidity than the healthy population. (See Revised manuscript)

9. Was the ability to read and write associated with medication adherence? Seems with the data at hand you could actually test the theory you are putting forward on lines 295-296.

Authors’ response: Thank you reviewer for the comment. We gave this possible explanation for why those who were unable to read and write had higher odds of poor sleep quality than who attained college and above based on our knowledge. As per your suggestion we have checked whether there was significant association between education and adherence using chi-square and odds ratio based on the cells in cross tabulation, which indicated that there was no significant association between them and we remove that statement in the discussion section. (See the revised manuscript)

10. Spelling mistake in title of final figure

Authors’ response: Thank you reviewer for the comment. We have modified it. (See the revised manuscript)

Reviewer#2

1. In the abstract and in line 87, replace "literature" by "authors" because it is the latter who suggest or show something. There is no need in the summary and discussion to represent the confidence interval. In line 42, replace "common" with important, great presence, any term that stands out more than common.

Authors’ response: Thank you for the comments, we have addressed all the comments. (See the revised manuscript)

2. In the introduction, line 58, improve sleep description. It is not "started" by a circadian cycle but has circadian rhythmicity. Furthermore, the correct one is the phase of wakefulness and not the period. Nor would I say that the quality of sleep "manifests" itself and the term "slumbering" is not very common in the literature.

Authors’ response: Thank you reviewer for the comment. We rephrase it and use the most appropriate definitions of sleep quality. (See the revised manuscript)

3. In line 66, is cited multitude of sequelae, I would like to see this paragraph more developed. The same applies for the consequences of poor sleep quality, on line 71.

Authors’ response: Thank you reviewer for the comments. We have addressed it. (See the revised manuscript)

4. I suggest paying attention to some parts of the work regarding the writing of English. For example, the paragraph in line 74 is a science communication text and not a text for a scientific journal at the PLOS ONE level. It should not be abandoned but rewritten with greater scientific rigor. There are also several small grammar errors throughout the text that a review by an English language specialist can be helpful.

Authors’ response: Thank you for the comments. We extensively modified the entire manuscript with the help of language experts. (See the revised manuscript)

5. In line 83, complete the study description reasoning. On line 91, remove "As to our search of the literature"

Authors’ response: Thank you reviewer for the comment. We have modified it. (See the revised manuscript)

6. I didn't understand the justification in lines 93, 94, 95. Elaborate this part better. On line 96, "Would be able" gets too vague. Need to be more assertive.

Authors’ response: Thank you reviewer for the comments. We have rewrote it. (See the revised manuscript)

7. In the results, in Table 1: Why age groups are not homogeneous: 18-24; 25-31; 32-38; 39-45; and > 45. I suggest using 5 groups.

Authors’ response: Thank you reviewer for the suggestions but we have categorizing the variable age based on previous literature and scientific evidence that is why the category is not homogeneous. Therefore, we keep it as it is.

8. In the body of the text, provide more details on the results shown in table 1, 2. Table 2 can be joined with table 3. Rethink the need to include figure 1, or include the other factors that are associated with poor sleep quality, "Being incapable of reading and writing", "undergoing polytherapy", "having poor medication adherence" and "having poor and moderate social support".

Authors’ response: Thank you reviewer for comments. The tables and figure have different information and we prefer it to present separately. (See the revised manuscript)

9. In table 4, it is not clear "<15mints + not during the last month" (?).

Authors’ response: Thank you for the comments, in the PSQI questionnaire tool component two was assessed as question #2 (after recoding was done ) sum with question #5a Score meaning that #2 Score (<15min (0), 16-30min (1), 31-60 min (2), >60min (3))+ #5a Score (if sum is equal 0=0; 1-2=1; 3-4=2; 5-6=3). 

10. Describe in the caption what IQR means. I see no need for Figure 2 unless the results are discussed in the later section.

Authors’ response: Thank you for the comments, we have addressed all the comments. (See the revised manuscript)

11. Remove the subheadings "Subjective sleep quality rate (component one)", "Sleep disturbance (component five)" and "Sleep quality prevalence" Provide a more detailed and joint description of all PSQI results.

Authors’ response: Thank you for the comments, we have addressed all the comments. (See the revised manuscript)

12. Do not repeat in line 272 what the objectives of the study are. Add the first paragraphs and remove any mention of standard deviations, confidence intervals, for example (95% CI; 64.8%, 72.5%) because we are in the discussion part and not the results part.

Authors’ response: Thank you for the comments, we have addressed all the comments. (See the revised manuscript)

13. The last paragraph reflects my criticism of the need to revise English. I understood what the authors mean but the text can improve a lot. It is necessary to discuss with references about the outcome of pain (component five). In the conclusions, cite what was said in the summary of the work about how to improve the quality of sleep in these patients.

Authors’ response: Thank you reviewer for the concerns. We used pain as one item to compute component five but not considered as a separate variable, then we aggregate the seven components to generate sleep quality global score. Then, based on the score we computed we categorized as poor or good. Therefore, we have not discuss about pain as it was not our objective/was not our independent variables. If these does not convince you we are ready to discuss about it, if it does not make our document lose its focus. (See the revised manuscript)

---

## [Editor Report · Decision Letter 1]

25 Nov 2021

PONE-D-21-26585R1Sleep quality and associated factors among adult patients with epilepsy attending follow-up care at Referral Hospitals in Amhara Region, EthiopiaPLOS ONE

Dear Dr. Simie,

Thank you for submitting your manuscript to PLOS ONE. After careful consideration, we feel that it has merit but does not fully meet PLOS ONE’s publication criteria as it currently stands. Therefore, we invite you to submit a revised version of the manuscript that addresses the points raised during the review process.

ACADEMIC EDITOR:I appreciate the authors' work to revise their manuscript. This version of manuscript is significantly improved compared to the last version. I have another minor comment which should be addressed before being published. The figure one is not necessary. So, please remove this figure. Instead, you can mention the frequency and frequency percent of these reasons in the text. For example: The most frequent reason for the difficulty in maintaining sleep was pain in XX participant (59.12%), followed by midnight wakeup in XX patients (48.32%),...

We look forward to receiving your revised manuscript.

Kind regards,

Hamidreza Karimi-Sari, MD

Academic Editor

PLOS ONE
---

## [Author Response · Author response to Decision Letter 1]

26 Nov 2021

Point by point response for editors/reviewers comments 

PLOS ONE Journal 

Manuscript title: Sleep quality and associated factors among adult patients with epilepsy attending follow-up care at Referral Hospitals in Amhara Region, Ethiopia

Manuscript ID: PONE-D-21-26585R1

Dear editor/reviewer. 

Dear all,

We would like to thank you for the constructive, building, and improvable comments on this manuscript that would improve the content of the manuscript. We considered each comment and clarification question of editors and reviewers on the manuscript thoroughly. Our point-by-point responses for each comment and question are described in detail on the following pages. Further, the details of changes were shown by track changes in the supplementary document attached.

Response to Editors comment

1. I appreciate the authors' work to revise their manuscript. This version of manuscript is significantly improved compared to the last version. I have another minor comment which should be addressed before being published. The figure one is not necessary. So, please remove this figure. Instead, you can mention the frequency and frequency percent of these reasons in the text. For example: The most frequent reason for the difficulty in maintaining sleep was pain in XX participant (59.12%), followed by midnight wakeup in XX patients (48.32%),...

Authors’ response: Thank you for the comments. As per your suggestion, we have removed Figure 1 and we wrote the frequencies with the corresponding percentage in text format. (See the revised manuscript)

---

## [Editor Report · Decision Letter 2]

1 Dec 2021

Sleep quality and associated factors among adult patients with epilepsy attending follow-up care at Referral Hospitals in Amhara Region, Ethiopia

PONE-D-21-26585R2

Dear Dr. Simie,

We’re pleased to inform you that your manuscript has been judged scientifically suitable for publication and will be formally accepted for publication once it meets all outstanding technical requirements.

Kind regards,

Hamidreza Karimi-Sari, MD

Academic Editor

PLOS ONE

---

## [Editor Report · Acceptance letter]

3 Dec 2021

PONE-D-21-26585R2 

Sleep quality and associated factors among adult patients with epilepsy attending follow-up care at Referral Hospitals in Amhara Region, Ethiopia 

Dear Dr. Simie Tsega:

I'm pleased to inform you that your manuscript has been deemed suitable for publication in PLOS ONE. Congratulations! Your manuscript is now with our production department. 

Kind regards, 

on behalf of

Dr. Hamidreza Karimi-Sari 

Academic Editor

PLOS ONE